# Antisense Oligonucleotide-Based Downregulation of the G56R Pathogenic Variant Causing *NR2E3*-Associated Autosomal Dominant Retinitis Pigmentosa

**DOI:** 10.3390/genes10050363

**Published:** 2019-05-10

**Authors:** Sarah Naessens, Laurien Ruysschaert, Steve Lefever, Frauke Coppieters, Elfride De Baere

**Affiliations:** 1Center for Medical Genetics Ghent, Ghent University and Ghent University Hospital, Corneel Heymanslaan 10, 9000 Ghent, Belgium; sarah.naessens@ugent.be (S.N.); laurien.ruysschaert@ugent.be (L.R.); steve.lefever@ugent.be (S.L.); frauke.coppieters@ugent.be (F.C.); 2Cancer Research Institute Ghent (CRIG), Ghent University, 9000 Ghent, Belgium; 3Bioinformatics Institute Ghent (BIG), Ghent University, 9000 Ghent, Belgium

**Keywords:** retinitis pigmentosa, autosomal dominant, *NR2E3*, G56R, putative dominant negative effect, gapmer antisense oligonucleotides, allele-specific knockdown

## Abstract

The recurrent missense variant in Nuclear Receptor Subfamily 2 Group E Member 3 (NR2E3), c.166G>A, p.(Gly56Arg) or G56R, underlies 1%–2% of cases with autosomal dominant retinitis pigmentosa (adRP), a frequent, genetically heterogeneous inherited retinal disease (IRD). The mutant NR2E3 protein has a presumed dominant negative effect (DNE) by competition for dimer formation with Cone-Rod Homeobox (CRX) but with abolishment of DNA binding, acting as a repressor in *trans*. Both the frequency and DNE of G56R make it an interesting target for allele-specific knock-down of the mutant allele using antisense oligonucleotides (AONs), an emerging therapeutic strategy for IRD. Here, we designed gapmer AONs with or without a locked nucleic acid modification at the site of the mutation, which were analyzed for potential off-target effects. Next, we overexpressed wild type (WT) or mutant *NR2E3* in RPE-1 cells, followed by AON treatment. Transcript and protein levels of WT and mutant NR2E3 were detected by reverse transcription quantitative polymerase chain reaction (RT-qPCR) and Western blot respectively. All AONs showed a general knock-down of mutant and WT NR2E3 on RNA and protein level, showing the accessibility of the region for AON-induced knockdown. Further modifications are needed however to increase allele-specificity. In conclusion, we propose the first proof-of-concept for AON-mediated silencing of a single nucleotide variation with a dominant negative effect as a therapeutic approach for *NR2E3*-associated adRP.

## 1. Introduction

Retinitis pigmentosa (RP, MIM 268000) encompasses a group of progressive inherited retinal diseases (IRDs) characterized by the loss of peripheral vision with the development of tunnel vison and subsequent loss of central vision [1]. Thirty to forty percent of all RP cases show autosomal dominant inheritance and currently 30 genes underlying autosomal dominant RP (adRP) have been identified [2,3]. One of them is the Nuclear Receptor Subfamily 2 Group E Member 3 (*NR2E3,* MIM 604485) gene, encoding a photoreceptor-specific nuclear receptor [4]. NR2E3 is part of a multi-protein complex with Neural Retina Leucine Zipper (NRL, MIM 162080) and Cone-Rod Homeobox (CRX, MIM 602225). This complex has a dual role in the development and function of rod photoreceptors by co-occupying the promoter and/or enhancer regions of photoreceptor-specific genes in the rod cells. In this way, NR2E3 suppresses cone gene expression and activates the expression of rod genes [5,6,7]. Mutations in *NR2E3* have been linked with autosomal recessive IRDs such as Enhanced S-cone syndrome (ESCS, MIM 268100) and Goldmann-Favre syndrome (GFS, MIM 268100). Furthermore, one specific founder variant in the DNA-binding domain of NR2E3, c.166G>A, p.(Gly56Arg) known as G56R has been found to cause 1%–2% of all adRP cases, being enriched in European populations and North American families of European descent [8,9,10,11]. This pathogenic variant is unique because of its presumed dominant negative effect (DNE) characterized by competition for dimer formation with CRX but loss of the necessary DNA-binding, which is caused by disruption of the α-helical DNA-binding motif [12,13] (Figure 1).

This DNE and the recurrence of the G56R mutation make it an interesting target for modulation with antisense oligonucleotides (AONs). AONs can bind their target RNA and recruit the RNAse-H1 enzyme, which degrades the RNA part of the formed RNA:DNA duplex and subsequently downregulates the target of interest [14]. Allele-specific knock-down using AONs or RNA interference has already been tested for various targets that exert a gain of function or dominant negative effect, such as *TMC1* (MIM 606706) for autosomal dominant hearing loss (MIM 606705) [15], *COL6A2* (MIM 120240), and *COL6A3* (MIM 120250) for Ullrich myopathy (MIM 254090) [16,17,18], *DNM2* (MIM 602378) for autosomal dominant centronuclear myopathy (MIM 160150 [19], *RHO* (MIM 180380) for adRP [20], *GCAP1* (MIM 600364) for adRP, and cone-rod dystrophy (MIM 602093 [21] and *HTT* (MIM 613004) for Huntington disease (MIM 143100) [22,23].

Recently, significant advances in AON-mediated therapy have been made in the field of IRD. For example, a recent clinical trial assessed intravitreal injections of an AON to restore correct splicing in ten patients with Leber Congenital Amaurosis (MIM 611755) due to a frequent deep-intronic mutation c.2991+1655A>G in *CEP290* (MIM 610142) [24]. This has a potential impact on other subtypes of IRD and other AON-mediated approaches are currently being investigated [25,26,27,28,29,30].

Here, we aimed to design and test the capacity of AONs to knock down the adRP-associated G56R mutation in an allele-specific manner, being one of the first studies employing allele-specific AONs to target a single nucleotide variation in IRD.

## 2. Materials and Methods

### 2.1. AON Design

A region of 150 bp around the c.166G>A mutation (Ensembl release 85) at the mRNA level was analyzed using Mfold software (version 3.6). Partially open or closed regions were identified using the ss-count tool. This led to the design of nine gapmer AONs overlapping with c.166G>A, differentiating in length of the DNA gap region, length of the RNA flanks, and the position of the mutation in the gap. Furthermore, we included the same nine gapmer AONs containing a locked nucleic acid (LNA) at the site of the mutation. Final AON sequences were analyzed using the OligoAnalyzer Tool (Integrated DNA Technologies, Coralville, IA, USA) to ensure that the GC content ranged between 40% and 60% and that the self-complementarity of the AON is above −9 kcal/mol. All nine AON pairs were analyzed for off-targets using Bowtie and Bowtie2 (Ensembl release 85), allowing a maximum of two mismatches.

All AONs were chemically modified by adding a phosphorothioate (PS) backbone to the complete AON sequence and 2-*O*-Methyl sugar modifications to the RNA flanks. AONs were generated by Eurogentec (Seraing, Belgium) and were dissolved in sterile TE-buffer to a final concentration of 100 µM.

### 2.2. Generation of Mutant c.166G>A Expression Construct

A wild type (WT) *NR2E3* (NM_014249) complementary DNA (cDNA) expression construct (OriGene, Rockville, MD, USA, TC308926) was used as a basis for mutagenesis. The c.166G>A mutation was inserted using the Q5 site-directed mutagenesis kit (New England Biolabs, Ipswich, MA, USA), and mutation-specific primers were designed with the NEBaseChanger software (version 1.2.8, New England Biolabs, Ipswich, MA, USA) (Appendix A). Mutagenized plasmids were transformed in One Shot TOP10 competent cells (Invitrogen, Life Technologies, Carlsbad, CA, USA) and DNA was subsequently isolated with the NucleoBond Xtra Midi/Maxi kit (Macherey-Nagel, Düren, Germany). The entire *NR2E3* insert was sequenced and selected plasmids were grown to obtain larger quantities. Sequencing primers can be found in Appendix A.

### 2.3. Cell Culture and AON Transfection Experiments

hTERT RPE-1 cells (ATCC, Manassas, VA, USA, CRL-4000™) were cultured in DMEM:F12 medium (ATCC, Manassas, VA, USA) supplemented with 10% fetal calf serum, 1% penicillin-streptomycin, 1% MEM non-essential amino acids, and 0.01 mg/mL hygromycin B. RPE-1 cells were seeded in six well plates at 1.5 × 10^5^ cells/well. After 24 h, cells were co-transfected in duplicate with 2 µg of WT or mutant *NR2E3* plasmid or an empty control vector and 0.25 µM of AON, using DharmaFECT kb DNA transfection reagent (Dharmacon, Lafayette, CO, USA) according to manufacturer’s instructions. Cells were grown for 48 h until confluence.

### 2.4. RNA Isolation and Reverse Transcription Quantitative Polymerase Chain Reaction (RT-qPCR)

Total RNA was extracted using the RNeasy mini kit (Qiagen, Hilden, Germany) with on-column DNase digestion (Qiagen, Hilden, Germany) according to manufacturer’s instructions. A second DNase treatment was done using the Heat&Run genomic DNA (gDNA) removal kit (ArcticZymes, Tromsø, Norway). cDNA was synthesized with the iScript cDNA synthesis kit (Bio-Rad Laboratories, Hercules, CA, USA) using 1 µg of messenger RNA (mRNA).

Reference genes that are stably expressed in RPE-1 cells were first determined using a GeNorm analysis, integrating the M and V analysis. The average pairwise variation M of a particular gene with all other control genes was determined as the standard deviation of the logarithmically transformed expression ratios. In addition, the systemic variation V was calculated as the pairwise variation for repeated RT-qPCR experiments on the same gene (qbase+, Biogazelle, Ghent, Belgium) [31]. The following reference genes were tested for stability: *HMBS*, *SDHA*, *HPRT1*, *PPIA*, *GUSB*, and *TBP*. Primer sequences are listed in Appendix A.

Next, the expression level of *NR2E3* was determined using intron-spanning primers (Primer3Plus, version 2.0). Data analysis was done using the qbase+ software (Biogazelle, Ghent, Belgium) with following reference genes for normalization: *HMBS*, *SDHA*, and *TBP.* Primer sequences are listed in Appendix A. Transfection and RT-qPCR were performed twice.

### 2.5. Western Blot Analysis

Cell lysates were collected in Leammli lysis buffer supplemented with complete protease inhibitor cocktail (Sigma-Aldrich, St. Louis, MO, USA) and phosphatase inhibitors (Sigma-Aldrich, St. Louis, MO, USA). Protein concentrations were measured using the Pierce BCA protein assay kit (Thermo Scientific, Waltham, MA, USA). A total of 25 µg of protein sample was reduced by incubation at 98 °C with 1M DTT. SDS-PAGE was performed using NuPAGE™ 4%–12% Bis-Tris protein gels (Invitrogen, Life Technologies, Carlsbad, CA, USA). Subsequently, proteins were transferred to a nitrocellulose membrane using the iBlot^®^2 System (Life Technologies, Carlsbad, CA, USA). Membranes were incubated in 2% membrane blocking agent (GE Healthcare Life Sciences, Chicage, IL, USA) in 1x tris-buffered saline with Tween-20 (TBST) for 2 h at room temperature and immunolabeled with a primary antibody (anti-NR2E3, #AB2299, Millipore, Burlington, MA, USA) diluted 1:1000 in blocking buffer for 16 h at 4 °C. Membranes were probed with a secondary antibody (anti-rabbit IgG, HRP-linked antibody #7074, Cell Signaling Technologies, Danvers, MA, USA) diluted 1:2500 in blocking buffer for 2 h at room temperature. Membranes were developed using the SuperSignal^®^ West Dura Extended Duration Substrate kit (Thermo Scientific, Life Technologies, Waltham, MA, USA) and scanned with the ChemiDoc-It^®^ 500 Imaging System (UVP, Upland, CA, USA). Subsequently, membranes were stripped using the Restore PLUS Western Blot Stripping Buffer (Life Technologies, Carlsbad, CA, USA) and probed with a β-tubulin primary antibody (ab6046, Abcam, Cambridge, UK) as an internal loading control. The antibody was diluted 1:2000 in blocking buffer and incubated for 1 h at room temperature. Membranes were subsequently probed with a secondary antibody (anti-rabbit IgG, HRP-linked antibody, Cell Signaling Technologies, Danvers, MA, USA) diluted 1:2500 in blocking buffer for 1 h at room temperature. The development was done as described above. Relative densities of NR2E3 were measured on three independent Western blots using Image J (version 1.8.0_112) and normalized against β-tubulin. Statistical analysis was done using an unpaired *t*-test.

## 3. Results

### 3.1. AON Design

We designed nine gapmer AONs targeting the c.166G>A mutation but shifted the oligonucleotides each time a few bps, as it is known that only a few bps difference can lead to a complete abolishment of inhibitory activity of AONs due to secondary or tertiary structures that restrict binding possibilities of the AON [32]. Furthermore, LNA modifications increase the difference in melting temperature between duplexes formed with a perfectly matched target versus a mismatched target and are thus possibly increasing the allelic-discrimination potency of AONs [33,34]. Hence, we included nine gapmer AONs containing a LNA modification at the site of the mutation. An overview of gapmer AONs around the site of the mutation, differentiating in length of the gap region, length of RNA flanks, and position of the mutation can be found in Table 1. Three pairs of gapmer AONs were finally selected that contain no complete match between the AON sequence and the full human gDNA sequence, and are further referred to as AON1–6 (Table 1). All three pairs have a 5-10-5 composition containing a DNA-gap region with 10 PS modified nucleotides, which is needed to allow cleavage of the RNAse-H1 enzyme. The gap region is flanked on both sides by 5 2′-*O*-Methyl PS RNA nucleotides.

### 3.2. NR2E3 mRNA Expression Analysis

All AONs were transfected in RPE-1 cells together with either the WT or mutant *NR2E3* cDNA expression construct to test whether the AONs induce selective silencing of the mutant NR2E3 mRNA and protein. For normalization of RT-qPCR expression data, we first determined reference genes that are stably expressed in RPE-1 cells using a GeNorm M and V analysis [31] testing following genes: *HMBS*, *SDHA*, *HPRT1*, *PPIA*, *GUSB*, and *TBP* (Appendix A). *HMBS*, *TBP*, and *SDHA* are the most stably expressed in RPE-1 cells (V < 0.15, lowest M values) and will be used in further experiments.

Next, relative quantities of WT and mutant *NR2E3* were determined using RT-qPCR of the harvested mRNA. All AONs show a non-selective knock-down of both WT and mutant *NR2E3* (Figure 2). However, when calculating the residual expression percentage after treatment relative to the non-treated sample, samples treated with AON2, 4, and 6 show lower remaining expression in the mutant versus the WT sample (Table 2). Interestingly, these AONs contain the LNA modification at the site of the mutation, possibly increasing the discrimination capacity of the AON [33].

### 3.3. NR2E3 Protein Expression Analysis

In parallel to mRNA expression analysis, Western blot analysis was performed to investigate whether the effects seen on mRNA level could be recapitulated on the protein level. A direct comparison of all conditions on the same gel was not possible, as the number of samples was too high. Therefore, a comparison of treated versus non-treated, and mutant versus WT samples was made per AON on the same gel. Three independent Western blots showed a clear partial knock-down of WT and mutant NR2E3 protein for all AONs (Figure 3A). Relative densities were calculated using β-tubulin for normalization. Unpaired *t*-tests between the non-treated sample and the treated samples revealed *p*-values between <0.0001 and 0.0328 (except for WT NR2E3 treated with AON4, *p*-value: 0.058) (Figure 3B). Transformation of these relative densities into percentages showed no clear difference in allele-specificity between AONs with or without LNA at the site of the mutation, as was seen for the mRNA expression. Instead, all AONs, except for AON2, seem to preferentially downregulate the mutant NR2E3, with AON5 being the most specific one (Table 3).

## 4. Discussion

We aimed to investigate if the adRP-associated NR2E3 mutation G56R with a dominant negative effect would be amenable to allele-specific knock-down using AONs. As other *NR2E3* mutations that have been found in autosomal recessive IRDs, such as enhanced S-cone syndrome (ESCS, MIM 268100), have a loss-of-function effect, we hypothesized that the c.166G>A allele could function as a therapeutic target for selective suppression [35] (Figure 1). Effective allele-selective approaches employing AONs have already been described for dominant mutations in *COL6A2* [18], *RHO* [20], and *HTT* [22]. However, gapmer AONs targeting a single nucleotide variation have not yet been described for IRD.

As *NR2E3* expression is mainly limited to the photoreceptors, we co-transfected WT or mutant expression constructs in RPE-1 cells, together with each of the six AONs targeting the c.166G>A allele. A general decrease in NR2E3 expression could be seen for all AONs both on the mRNA and protein level, but not limited to the mutant NR2E3. However, on the mRNA level, a subtle mutant-specific preference could be seen for AON2, 4, and 6. Interestingly, these contain the LNA modification at the site of the mutation. On the protein level, a preferential mutant-specific knock-down could be observed for all AONs except for AON2 in three independent experiments. Apart from the fact that Western blot is a semi-quantitative technique, the observed mRNA-protein differences may be attributed by other factors, e.g., different gapmer AON activities in the nucleus versus cytoplasm [18].

The six evaluated AONs downregulate NR2E3 expression to 28%–60% at the mRNA level and to 9%–70% at the protein level. Importantly, in the context of the dominant negative *NR2E3* mutation, a complete AON-induced knock-down of the mutant allele is not required. Morphological or cellular improvements resulting from moderate reductions in mutant protein while preserving the WT protein [20] or from shifting the WT-mutant protein ratio [16,17] have been described respectively. Gualandi et al. showed that AON-targeting of a common single nucleotide polymorphism (SNP) caused a global suppression of *COL6A2* mRNA, albeit preferentially involving the mutated allele (10% difference in mutant versus WT allele ratio). Overall, this led to a significant improvement of the severe cellular phenotype in patient fibroblasts [16].

In order to increase the allele-specificity of the AONs targeting *NR2E3* mutation G56R, both the cellular system and AON sequences could be modified. As we had no access to patient-derived induced pluripotent stem cells (iPSCs) differentiated to photoreceptor precursor cells (PPCs) at the start of the study, we used an overexpression cellular system either for mutant or WT *NR2E3*. This is not representing the true biological situation however, in which both mutant and WT NR2E3 are expressed, exerting a dominant negative effect. In *NR2E3* expressing patient-derived cells, competition for AON binding might lead to a stronger allele-specificity of the AON. In this respect, iPSCs have recently been reprogrammed from patient fibroblasts with the c.166G>A mutation [36]. These cells will further be differentiated to PPCs [37], which will represent a relevant cellular model for further AON development.

Apart from using a different cellular system, several modifications to the AONs could be made to obtain a more selective knock-down of the mutant allele. A first potential modification could be shifting the AONs a few bps. Already three positions of AONs targeting the mutation have been investigated here, but several studies showed that even a change of one bp can have a substantial influence on allelic discrimination [38,39]. Next, reducing the length of the AONs could have positive effects on the discrimination capacity. The AONs used in this study have a 5-10-5 conformation. However, it has been shown that a long AON sequence can reduce the allele-specificity [18]. Shortening the AONs to 16 nucleotides could increase specificity on the one hand [39,40], but lead to an increase of possible off-target effects on the other hand (see Table 1). Finally, it has been suggested that the discrimination of single nucleotides between RNA alleles can be improved by introducing certain arrangements of mismatches between the RNA/AON duplexes, as the presence of non-canonical bps decreases the thermodynamic stability of the duplexes. In case of a G>A transition, it has been shown that the presence of G-dT in the WT RNA duplex instead of A-dT does not disturb the duplex structure. However, introducing tandem purine mismatches at the 5′ end of the DNA gap strongly diminishes thermodynamic stability. This, combined with the third mismatch at the site of the mutation, has been shown to reduce the cleavage efficiency of the WT RNA [41].

In conclusion, we have designed and investigated gapmer AONs to elicit allele-specific silencing of a recurrent dominant negative mutation in the *NR2E3* gene implicated in 1%–2% of cases with adRP. We showed that the region of interest is accessible to AON-induced RNAse-H1 cleavage and that AON-targeting of G56R caused a global suppression of NR2E3 mRNA and protein, with limited mutation-specificity. Further studies will evaluate different modifications to AONs, and will validate gapmer AONs in a relevant patient-derived cellular model. Finally, our findings provide the first proof-of-concept for AON-mediated silencing of a single nucleotide variation with a dominant negative effect as a therapeutic approach for dominant IRD.

## Figures and Tables

**Figure 1 genes-10-00363-f001:**
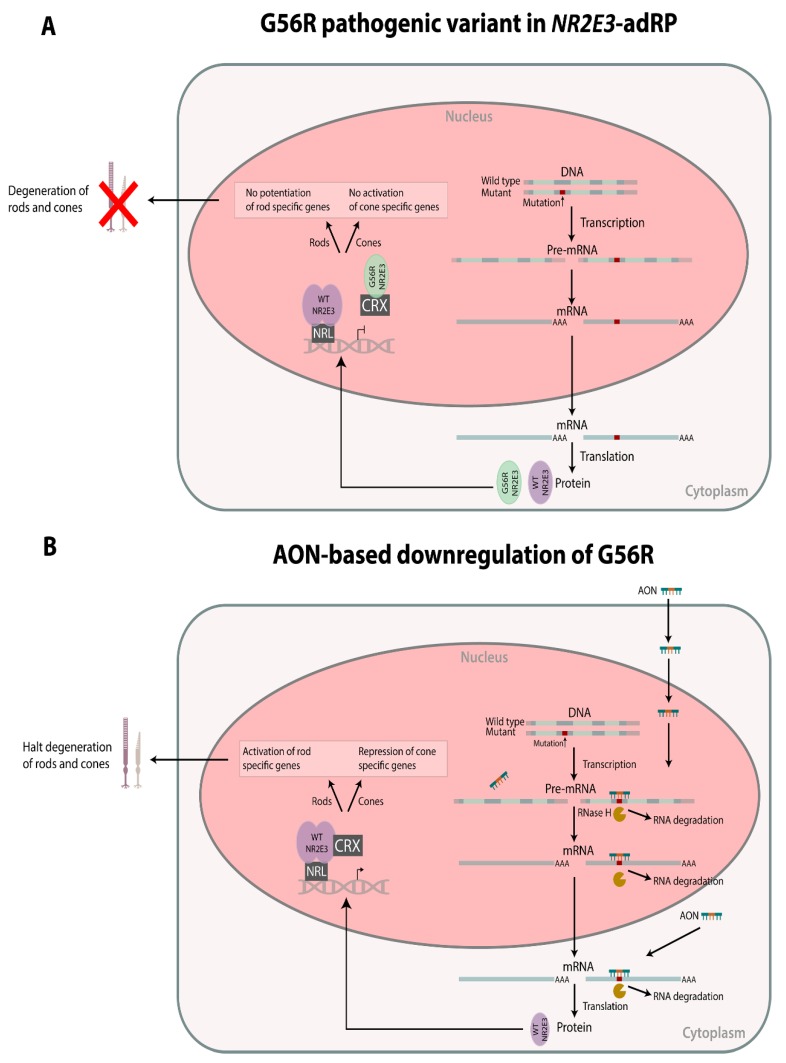
Schematic overview of the dominant negative effect of G56R Nuclear Receptor Subfamily 2 Group E Member 3 (NR2E3) (**A**) and allele-specific silencing of G56R using antisense oligonucleotides (AONs) (**B**). (**A**) Both wild type (WT) and mutant alleles are transcribed and translated. The wild type protein is able to form homo-dimers and bind the DNA, in complex with Neural Retina Leucine Zipper (NRL). The mutant protein is competing for binding with Cone-Rod Homeobox (CRX) and this complex is no longer able to bind the DNA, which leads to a failure of both potentiation of rod specific genes and activation of cone specific genes. (**B**) Using AONs, the mutant allele can be selectively downregulated, whereby the WT protein is binding the DNA in complex with both NRL and CRX. This complex is able to properly activate the rod specific genes and repress the cone specific genes, which would halt degeneration of the photoreceptors. Adapted from [18].

**Figure 2 genes-10-00363-f002:**
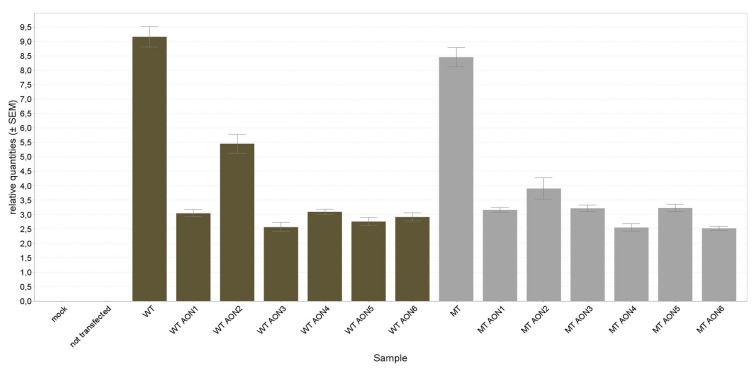
**Quantitative polymerase chain reaction** (qPCR)-based expression analysis of messenger RNA (mRNA) from wild type (WT) or mutant (MT) *NR2E3* overexpressed in RPE-1 cells and subsequently treated with AON1–6. All AONs lead to a partial knockdown of WT or MT *NR2E3* mRNA expression. Error bars represent the standard error of the relative quantities. Mock transfected cells were transfected with an empty vector and show no *NR2E3* expression.

**Figure 3 genes-10-00363-f003:**
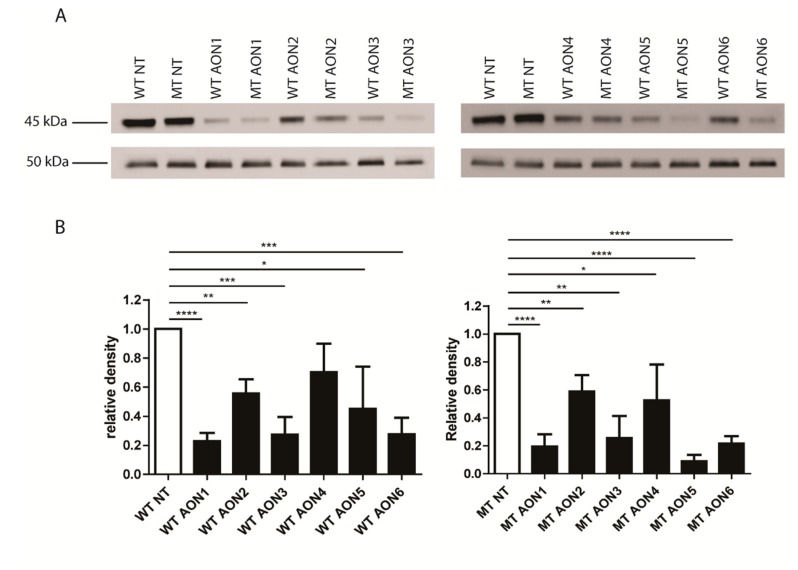
Western blot analysis. Wild type (WT) or mutant (MT) NR2E3 was overexpressed in RPE-1 cells and subsequently treated with AON1–6. (**A**). Upper images represent NR2E3 protein visualization, lower images represent β-tubulin visualization. (**B**). Densitometric analysis for three independent Western blots. Left panel represents WT samples, right panel represents MT samples. All AONs lead to a partial knock-down of WT and MT NR2E3 protein (*p*-values between <0.0001 and 0.0328). Error bars represent standard deviation between relative densities.

**Table 1 genes-10-00363-t001:** Antisense oligonucleotide (AON) sequences and potential off targets. Underlined AONs were selected, as they do not have predicted off targets at the genomic DNA (gDNA) level. cDNA = complementary DNA, M = 2-*O*-Methyl modification, L = locked nucleic acid modification, * = phosphodiesterase bond. The site of the mutation is indicated with a capital letter.

Gapmers	Sequence (5′→3′)	Complete Match Bowtie	Complete Match Bowtie2
AON1 AON2	Mg*Mu*Mg*Mc*Mu*t*c*c*T*g*c*t*g*c*t*Mg*Mc*Mu*Mg*Mu Mg*Mu*Mg*Mc*Mu*t*c*c*LT*g*c*t*g*c*t*Mg*Mc*Mu*Mg*Mu	gDNA: no cDNA: *SLC29A4P1*	gDNA: no cDNA: no
	Mg*Mc*Mu*t*c*c*T*g*c*t*g*c*t*Mg*Mc*Mu Mg*Mc*Mu*t*c*c*LT*g*c*t*g*c*t*Mg*Mc*Mu	gDNA: *ICOS* cDNA: *GPR27*	gDNA: *CACNA2D3* cDNA: *PCDHAC2*
	Mg*Mu*Mg*Mc*Mu*t*c*c*T*g*c*t*g*Mc*Mu*Mg*Mc*Mu Mg*Mu*Mg*Mc*Mu*t*c*c*LT*g*c*t*g*Mc*Mu*Mg*Mc*Mu	gDNA: intergenicc DNA: *HCG9P5*	gDNA: intergenicc DNA: *PRDM8*
AON3 AON4	Mc*Ma*Mu*Ma*Mg*t*g*c*t*t*c*c*T*g*c*Mu*Mg*Mc*Mu*Mg Mc*Ma*Mu*Ma*Mg*t*g*c*t*t*c*c*LT*g*c*Mu*Mg*Mc*Mu*Mg	gDNA: no cDNA: no	gDNA: no cDNA: no
	Mu*Ma*Mg*t*g*c*t*t*c*c*T*g*c*Mu*Mg*Mc Mu*Ma*Mg*t*g*c*t*t*c*c*LT*g*c*Mu*Mg*Mc	gDNA: intergenicc DNA: *HCG9P5*	gDNA: *TANGO6* cDNA: *HCG9P5*
	Ma*Mu*Ma*Mg*Mu*g*c*t*t*c*c*T*g*Mc*Mu*Mg*Mc*Mu Ma*Mu*Ma*Mg*Mu*g*c*t*t*c*c*LT*g*Mc*Mu*Mg*Mc*Mu	gDNA: *AC007908.1* cDNA: no	gDNA: no cDNA: *HCG9P5*
AON5 AON6	Mc*Mu*Mu*Mc*Mc*T*g*c*t*g*c*t*g*c*t*Mg*Mu*Mc*Mu*Mc Mc*Mu*Mu*Mc*Mc*LT*g*c*t*g*c*t*g*c*t*Mg*Mu*Mc*Mu*Mc	gDNA: no cDNA: *PCDHAC2*	gDNA: no cDNA: *ACVR2B*
	Mu*Mu*Mc*c*T*g*c*t*g*c*t*g*c*Mu*Mg*Mu Mu*Mu*Mc*c*LT*g*c*t*g*c*t*g*c*Mu*Mg*Mu	gDNA: *EIF3H* cDNA: *LMAN2*	gDNA: *PDCD1LG2* cDNA: *DYRK1A*
	Mc*Mu*Mu*Mc*Mc*T*g*c*t*g*c*t*g*Mc*Mu*Mg*Mu*Mc Mc*Mu*Mu*Mc*Mc*LT*g*c*t*g*c*t*g*Mc*Mu*Mg*Mu*Mc	gDNA: *SEMA5A* cDNA: *LY6H*	gDNA: *SEMA5A* cDNA: *PLXNB3*

**Table 2 genes-10-00363-t002:** Relative expression values of messenger RNA (mRNA) from wild type (WT) or mutant (MT) *NR2E3* overexpressed in RPE-1 cells and subsequently treated with AON1–6. Residual expression after AON treatment was calculated relative to the non-treated (NT) sample (=100%). SEM = standard error of relative expression values.

AON	Expression WT *NR2E3* (%)	SEM (%)	Expression MT *NR2E3* (%)	SEM (%)
NT	100.00	3.84	100.00	4.01
AON1	33.22	4.06	37.37	2.86
AON2	59.55	6.12	46.15	9.65
AON3	27.98	5.98	38.04	3.51
AON4	33.76	2.94	30.15	4.89
AON5	30.10	4.95	38.16	3.87
AON6	31.81	4.94	29.89	4.26

**Table 3 genes-10-00363-t003:** Relative protein expression values from wild type (WT) or mutant (MT) NR2E3 overexpressed in RPE-1 cells and subsequently treated with AON1–6. Average residual protein expression after treatment with AON1–6 was calculated from three independent blots relative to the non-treated (NT) sample (=100%).

AON	Expression WT NR2E3 (%)	Expression MT NR2E3 (%)
NT	100.00	100.00
AON1	22.87	19.33
AON2	55.55	58.85
AON3	27.22	25.45
AON4	70.24	52.47
AON5	45.08	8.88
AON6	27.52	21.7

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
