# Peer review of "Antisense Oligonucleotide-Based Downregulation of the G56R Pathogenic Variant Causing NR2E3-Associated Autosomal Dominant Retinitis Pigmentosa"

_genes, 2019, doi:10.3390/genes10050363_

Round 1

Reviewer 1 Report

General Remarks

Very interesting and comprehensive article. The science appears to be well executed and well understood. The conclusions drawn also illustrate a broad and in-depth knowledge of the field. 

Abstract

Lines 16-18: This sentence could be rephrased to better depict the relationship of the proteins involved. For example, would the mutant protein be better described as an antagonist/inverse agonist? Is wild-type binding disrupted due to conformational changes, steric hindrance or something else?

Introduction

Line 33: The use of MIM identifiers is inconsistent through out the article. They should be at least included for key genes and conditions that are sufficiently described/relevant to the subject.

Lines 38-41: Consider deconstructing this sentence as it currently uses "and" five times.

Line 39: NR1D1 is described as a noteworthy component of the NR2E3 multi-protein complex yet does not appear in Figure 1. For consistency, I would recommend omitting NR1D1 from this sentence as it may be difficult to incorporate into the diagram given the complexity of the relationship.

Line 44: If using the term "founder variant", some information relating to the founder event should be provided. Perhaps mentioning the cause of the founder effect or the geographic region where it is now enriched.

Lines 46-47: These lines mirror lines 16-18 in the Abstract, again some clarity is required. 

Lines 60-61: The language in this sentence is inconsistent with the tone of the rest of the article, consider rephrasing "high hopes".

Methods

Line 113: Change "references genes" to "reference genes". 

Line 114: Change "Following reference genes" to "The following reference genes".

Line 120: Change "was" to "were".

Results

Very convincing data, well done.

Discussion

Line 220: This statement regarding expression should be double-checked. There are some expression studies from GTEx and BioGPS that suggest NR2E3 expression is not limited to the retina. However, I do not believe that this should have any downstream implications for AON therapies targeting the retina that are delivered directly to the eye.

Lines 244-248: It may also be valuable to evaluate the expression levels of the other key components of the NR2E3 complex (CRX,NRL,NR1D1) if such a resource will be available. Additionally, these cells could also be used to analyse the potential off-target effects identified by Bowtie/Bowtie2. This is a suggestion only, no in-text changes are required.

Reviewer 2 Report

Naessens et al. present a very well written concise report on proof-of-concept data  for AON-based therapy for NR2E3-p.G56R-linked ADRP. The experiments are carefully designed, with appropriate controls, and clearly presented. The results are not spectacular, but are important. Publishing it in Genes appears to be appropriate to this reviewer.

Minor comments:

Abstract, Introduction and Figure 1: NR2E3 is able to bind as a homodimer directly to binding sites on the DNA, and NRL may bind in complex with NR2E3 to adjacent response elements. The dominant negative effect of DNA-binding defective NR2E3-p.G56R describes actually the fact that the NR2E3 dimer formed by wild-type and mutant protein is no more able to bind DNA. The hypothesized effect of NR2E3-p.G56R hindering binding of CRX to its binding sites is a slightly differnt mechanism.  

Please explain the GeNorm analysis in the MatMeth section. This reviewer understands why HMBS, TBP and SDHA were used for normalization, but not the general approach of this V and M analysis

Typos:

- unify Fonts in lines 115-116

- the authorlist of ref 12 in line 318 is not correct
